# Holocene Climate Dynamics in the Central Mediterranean Inferred from Pollen Data

Léa d'Oliveira<sup>1</sup>, Sébastien Joannin<sup>1</sup>, Guillemette Ménot<sup>2</sup>, Nathalie Combourieu-Nebout<sup>3</sup>, Lucas Dugerdil<sup>1,2</sup>, Marion Blache<sup>1,4</sup>, Mary Robles<sup>5</sup>, Assunta Florenzano<sup>6</sup>, Alessia Masi<sup>7</sup>, Anna Maria Mercuri<sup>6</sup>, Laura Sadori<sup>7</sup>, Marie Balasse<sup>8</sup>, and Odile Peyron<sup>1</sup>

Correspondence: Léa d'Oliveira (lea.d-oliveira@umontpellier.fr)

Abstract. The Mediterranean climate is characterised by strong seasonality, which is critical for the ecosystems and societies in the region and makes them susceptible to climate change. The timing of when the Mediterranean climate de-5 veloped over the past few thousand years remains a complex and unresolved question. Most studies document a part of the Mediterranean area or are based on a single (and frequently different) climate reconstruction method, which can lead to non-negligible biases when considering climate changes 10 on a Mediterranean scale. Several climate summaries based on pollen data have recently been produced on a European scale. However, few of them have focused exclusively on the Mediterranean area, except for two recent syntheses documenting the eastern and western parts of the Mediterranean 15 basin. We aimed to document the climate changes in the central Mediterranean during the Holocene, including trends and different patterns. A robust methodology has been applied to 38 pollen records spreading across the south of France and Italy. Four climate reconstruction methods based on differ-20 ent mathematical and ecological concepts have been tested (MAT, WA-PLS, BRT and RF), and the selection of the best modern calibration dataset has also been investigated to produce the most reliable results. Particular attention has been paid to the seasonal nature of climatic parameters (winter and 25 summer temperatures and precipitation). A model-data comparison has been made using the transient model simulation TraCE-21ka in an attempt to gain a better understanding of the climate mechanisms and their forcing. Our palaeoclimate reconstruction shows that during the mid Holocene, summer temperatures were slightly colder than modern-day condi- 30 tions in the southern part of the central Mediterranean region, which is not completely in accordance with the summer temperature reconstructions of the Iberian Peninsula and eastern Mediterranean for the mid Holocene. In northern parts of the central Mediterranean region, and particularly in high elevation (> 1000 m), a Holocene thermal maximum is present, contrasting with the cold summer temperature anomalies previously reconstructed with pollen data for the Mediterranean region. Holocene summer conditions were characterised by specific spatio-temporal patterns, i.e., a west-east differen- 40 tiation in southern France and a north-south one in Italy, for both temperature and precipitation. Holocene winter conditions showed a more homogeneous spatio-temporal pattern, i.e., general humidification and warming throughout the Holocene for Italy and southern France, which is co- 45 herent with the winter temperature reconstructions of the Iberian Peninsula and eastern Mediterranean. A data-model comparison shows a mostly coherent signal in winter but an incoherent one in summer. Those discrepancies between model simulations and pollen-based reconstructions suggest 50 that during the Holocene, the northern Mediterranean climate was already subject to a marked spatio-temporal variabil-

<sup>&</sup>lt;sup>1</sup>Université de Montpellier, CNRS, IRD, EPHE, UMR 5554 ISEM, 34090, Montpellier, France

<sup>&</sup>lt;sup>2</sup>Univ. Lyon, ENS de Lyon, Université Lyon 1, CNRS, UMR 5276 LGL-TPE, 69364, Lyon, France

<sup>&</sup>lt;sup>3</sup>Muséum national d'Histoire naturelle, CNRS, MNHN, UMR 7194 HNPH, 75116, Paris, France

<sup>&</sup>lt;sup>4</sup>Université du Québec en Abitibi-Témiscamingue, IRF, Québec, Canada

<sup>&</sup>lt;sup>5</sup>Université Aix Marseille, CNRS, IRD, INRAE, Collège de France, UMR 7330 CEREGE, 13545, Aix-en-Provence, France

<sup>&</sup>lt;sup>6</sup>University of Modena and Reggio Emilia, LPP, 41121, Modena, Italy

<sup>&</sup>lt;sup>7</sup>Sapienza University of Rome, DBA, 00185, Rome, Italy

<sup>&</sup>lt;sup>8</sup>Muséum national d'Histoire naturelle, CNRS, MNHN, UMR 7209 BioArch, 75116, Paris, France

ity, particularly in summer, that cannot only be explained by changes in orbital configuration and atmospheric greenhouse gas evolution. Finally, our result highlighted the onset of the "Mediterraneanization" of the central Mediterranean region, characterised by wet winters and dry summers, after 8,000 years Before Present (BP). The "Mediterraneanization" process seems to have had a greater impact on the southern regions than on the northern regions.

### 1 Introduction

10 Placing recent global warming in the context of natural climate variability requires a long-term perspective. Climate changes during the Holocene (the last 11,700 years BP) have been extensively documented by various palaeoclimate proxy records, revealing complex and heterogeneous spatio-15 temporal patterns in Europe (e.g., Cheddadi et al., 1997; Davis et al., 2003; Magny et al., 2003; Bartlein et al., 2011; Kaufman et al., 2020b; Herzschuh et al., 2023a). These climate patterns may be linked to regional characteristics (e.g., latitude or elevation) or seasonal variables (e.g., summer 20 or winter) (Mauri et al., 2015; Cartapanis et al., 2022; Erb et al., 2022), which can in turn impact how the pattern is recorded. This effect is well illustrated by the temperature warming between 10,000 and 6,000 years BP, also called mid Holocene thermal maximum (HTM), well evidenced in 25 northern Europe (Renssen et al., 2012; Liu et al., 2014) but poorly recorded in southern Europe (Kaufman et al., 2020b; Marriner et al., 2022).

In the Mediterranean area, terrestrial proxies suggest conditions similar to or cooler than those observed today dur-30 ing the mid-Holocene period (Cheddadi et al., 1997; Davis et al., 2003; Herzschuh et al., 2023a). However, over the last decades, atmospheric climate models often fail to simulate the mid Holocene Mediterranean cooling, producing significant warming both in northern and southern Europe 35 (Liu et al., 2014; Mauri et al., 2014; Erb et al., 2022). Recent model studies are in better agreement with the data as they integrate the soil-atmosphere interactions in their model simulations (Russo and Cubasch, 2016; Russo et al., 2022), but the underlying climate mechanisms are not yet fully un-40 derstood. This is still a key question, as most data or model studies focus on the European region or the world as a whole (Cheddadi et al., 1997; Davis et al., 2003; Mauri et al., 2015; Marsicek et al., 2018; Kaufman et al., 2020b; Cartapanis et al., 2022; Erb et al., 2022; Herzschuh et al., 2023a), but few 45 on the Mediterranean region, which is in the end not enough studied on a broad scale (Magny et al., 2012; Peyron et al., 2017; Finné et al., 2019; Marriner et al., 2022).

The Mediterranean region is located in a complex transitional zone, influenced by both the tropical circulation cells and the mid-latitude westerlies and cyclogenesis, which exposes the basin to a relatively large spectrum of climatic influences from the arid zone of the subtropical high to the hu-

mid north-westerly air flows (Margaritelli et al., 2020). In summer, the Mediterranean region is under the influence of a subtropical climate, which is strongly linked to the Inter-Tropical Convergence Zone (ITCZ) position. In winter, however, the subtropical high-pressure belt is shifted southward and the Mediterranean region is mainly linked to the westerly system bringing an eastward important water influx (Giraudi et al., 2011; Goudeau et al., 2015). This results in a pronounced rainfall seasonality that is critical for ecosystems and societies in the Mediterranean region today and in the past (Cramer et al., 2018) and points out the need for seasonal scale reconstructions over annual ones (Goudeau et al., 2015).

Few studies have attempted to better understand the climate changes at a regional scale in the Mediterranean region during the Holocene (Jalut et al., 2000; Magny et al., 2013; Peyron et al., 2017; Finné et al., 2019). Documenting these climate changes with precision is complex, as the 70 reconstructed climate patterns may be linked to regional characteristics (e.g., longitude, latitude, and elevation), seasonal variables (e.g., summer or winter), or specific to the proxy used (pollen, chironomids, marine proxies, molecular biomarkers). Specific temperature and precipitation patterns have been suggested as a west-east (Dormoy et al., 2009; Liu et al., 2023) as well as a north–south gradient within the Mediterranean basin (Magny et al., 2012, 2013; Peyron et al., 2017). These studies concluded that, over the past 10,000 years, the Mediterranean basin has not been defined by a 80 singular climate trajectory, making our understanding of the spatio-temporal variability of the Mediterranean climate still incomplete. Similarly, the intensity of seasonal variations, e.g., the differences between summer and winter conditions, and their evolution through time also seem to vary greatly 85 from one region to another during the Holocene (Roberts et al., 2011; Peyron et al., 2017).

Climatic reconstructions heavily rely on the proxies selected (Samartin et al., 2017; Kaufman et al., 2020a; Cartapanis et al., 2022). Notably, chironomid-based reconstructions disagree with those derived from pollen data (Samartin et al., 2017; Cartapanis et al., 2022). In the Mediterranean region, a multi-proxy approach is commonly employed to examine past climate changes, integrating terrestrial and marine records (lake levels, fluvial activity, pollen records, Ja- 95 lut, 2000; Roberts et al., 2011; Finné et al., 2019; Cartapanis et al., 2022; Marriner et al., 2022) as well as only pollenbased approaches (Davis et al., 2003; Mauri et al., 2015; Peyron et al., 2017; Cruz-Silva et al., 2023; Liu et al., 2023). Pollen is favoured in palaeoclimate research due to its wide 100 spatial coverage and the well-established connection between vegetation and climate, which enhances the reliability of these reconstructions. However, discrepancies arise from using different reconstruction methods (transfer functions), varying pollen datasets, and diverse climatic (e.g., precipita- 105 tion, temperature) and bioclimatic (e.g., growing degree day, evapotranspiration) parameters, which can affect result accuracy (Chevalier et al., 2020). Recent regional analyses utilising high-quality pollen data have yielded insights into climate changes in both the western Mediterranean (brown dots on Fig. 1, Liu et al., 2023) and the eastern Mediterranean (blue dots on Fig. 1, Cruz-Silva et al., 2023). These studies revealed complex spatio-temporal changes across these key Mediterranean regions and emphasised the existing knowledge gap regarding the intermediate sub-regions of the north-central Mediterranean, particularly southern France and Italy.

In this frame, our study aims to reconstruct the spatio-temporal climate variability of this central region to better understand and discuss the climate changes at the scale of the Mediterranean area. Our questions are: (1) What were the climate conditions during the mid Holocene? Should we recon-

mate conditions during the mid Holocene? Should we reconstruct colder than today's conditions, as in previous pollenbased studies? Is the HTM recorded in the Mediterranean
area during the early to mid Holocene, as shown by marine
proxies? (2) Is there a north-south climate pattern across the
Italian peninsula and adjacent regions and an east-west at the
Mediterranean scale during the Holocene? (3) Is it possible
to document, with accuracy, from pollen data, the seasonality
of climate? In particular, the summer and winter parameters,
as many studies diverge on the summer signal. Are the climate models (GCMs and regional) simulations in agreement
with our results? (4) When and how did the Mediterranean
conditions we know today come about?

In contrast to previous synthesis (Cheddadi et al., 1997; Davis et al., 2003; Mauri et al., 2014, 2015; Cruz-Silva et al., 2023; Liu et al., 2023), our study is based on a multi-method 30 approach (e.g., Peyron et al., 2005; Brewer et al., 2008; Peyron et al., 2011, 2013, 2017; Salonen et al., 2019; Robles et al., 2022; d'Oliveira et al., 2023; Robles et al., 2023) applied to 38 fossil pollen records located in the north-central Mediterranean. Such an approach is more reliable than stud-35 ies based on a single method, such as Modern Analogue Technique (MAT) or Weighted Averaged Partial Squared (WA-PLS) methods (Chevalier et al., 2020). We will give particular attention to document the seasonality of temperature and precipitation interpretations because climate forcing 40 during the Holocene was dominated by orbitally controlled insolation changes that operated asymmetrically across the annual cycle (Renssen et al., 2012). To increase the reliability of our reconstruction, a first stage of numerical tests (autocorrelation, canonical correspondence analysis) will be carried 45 out on three modern climate and pollen datasets (global and regional) for annual and seasonal (spring, summer, autumn and winter) temperature and precipitation parameters.

Our results will be compared (1) to transient model simulations (TraCE-21ka) from the Community Climate System Model (CCSM3; Collins et al., 2006) and (2) to the synthesis of Liu et al. (2023) for the Western Mediterranean and those of Cruz-Silva et al. (2023) for the Eastern Mediterranean to better understand the climate changes at the Mediterranean scale. The data-models comparison will give us a better unses derstanding of climate mechanisms and their forcings.

### 2 Material and methods

### 2.1 Study area

The study region covers the north-central Mediterranean Basin (Fig. 2a) and is divided into two study zones, i.e., southern France and Italy.

Zone 1 (southern France) extends from the southern French coast to the north of the Massif Central, including part of the Pyrenees and the Alps and extends from longitude 1° E to longitude 8° E. This region is influenced by a Mediterranean climate, associated with cool mild winters 65 and hot dry summers (Fig. 2b-c). Mean annual precipitation ranges from 500 to 800 mm, with a maximum recorded in autumn and a minimum in summer. Rainfall seasonality is influenced by altitude, resulting in an increase in spring precipitation with altitude while temperatures decrease (Azuara et al., 2020). Because of its latitude, southern France is directly influenced by large-scale atmospheric circulation patterns over the North Atlantic and Europe, i.e., a persistent low-pressure area located over Iceland, alongside the semipermanent high-pressure system associated with the Azores 75 (Brandimarte et al., 2011). Low-pressure conditions over Iceland, associated with an anticyclonic pattern in the eastern Atlantic, result in cold, dry, northerly winds over southern France, particularly in the Gulf of Lion. Meanwhile, high pressure brings warm, humid, southeasterly winds in the 80 region (Azuara et al., 2020). This dynamic interaction between Iceland and the Azores influences westerly airflow patterns, affecting thermal exchanges across the North Atlantic and Europe, especially during the winter months at mid to high latitudes. The westerly flow also varies interannually, shifting between northern and southern paths. This variation, known as the North Atlantic Oscillation (NAO), plays a significant role in climate variability in the western Mediterranean. The strength of the NAO is determined by the difference in normalised sea level pressure (SLP) between high and low latitudes in the Atlantic (Azuara et al., 2020). A positive NAO indicates a stronger meridional pressure gradient and more intense westerly winds, while a negative NAO suggests a weaker gradient and weaker westerlies. During winters with a positive NAO, subtropical atmo- 95 spheric pressure tends to rise, while Arctic pressure drops. This setup leads to varying winter weather: southern Europe experiences warmer and drier conditions, northern Europe sees warmer and wetter weather, and Greenland faces colder, drier winters (Brandimarte et al., 2011).

Zone 2, corresponding to Italy, is located in the northern-central Mediterranean basin and stretches from latitude 36° N to latitude 47° N, making for contrasting climatic conditions from one region to another. The Italian orography is also complex and contrasted due to the presence of the Alps to the north and the Apennines, which stretch along the entire peninsula, acting as a barrier to air masses from continental Europe. Both mountain chains influence the pathway of

Figure 1. Spatial coverage of pollen-based Holocene climate syntheses for the northern Mediterranean basin focusing on the Iberian Peninsula (Liu et al., 2023), the Eastern Mediterranean (Cruz-Silva et al., 2023) and the north-central Mediterranean (this study).

weather fronts and interact with dominant winds, thereby exposing various regions of Italy to distinct circulation patterns (Fratianni and Acquaotta, 2017). The Alps act as a barrier, limiting the influx of cold air masses from central Europe 5 into the Po Valley and northern Italy. The Apennines diminish the vulnerability of western areas to cold easterly winds that originate from the Balkans, while the Adriatic Sea, situated on the upwind side, is more susceptible to strong winds and heavy rainfall (Di Bernardino et al., 2024). The Italian 10 Peninsula, surrounded by the Mediterranean Sea to the west, south and east, has its climate strongly mitigated by the presence of the water body, which represents an important source of heat and moisture (Winschall et al., 2014). The simultaneous impact of multiple geographic elements (e.g., elevation, 15 distance from the sea, presence of particular coastal currents, exposure to dominant winds, etc.) shapes the presence of distinct climatic zones, each associated with particular prominence of weather types.

### 2.2 Pollen datasets

### 20 2.2.1 Fossil pollen data

Here, 38 pollen records (Tab. 1) were selected and extracted using the international database Neotoma (Williams et al., 2018) or gathered directly from the authors of the original studies. These records were selected according to specific criteria such as location, temporal extent and resolution. The aim was to compile the most suitable records for synthesis-

ing the Mediterranean climate of the Holocene (Fig. 2a). The selection procedure followed the following steps:

- Location filter: Records were restricted to southern France (latitude 42–45° N; longitude 1–8° E) and Italy (latitude 37–47° N; longitude 8–16° E), yielding 749 pollen records from Neotoma.
- Age-depth model update: LegacyAge 1.0 dataset (Li et al., 2022) was used to update the age-depth models of each record. The LegacyPollen database, com- 35 piled by Herzschuh et al. (2022), contains 2,831 pollen records from all over Europe, covering the last 30,000 years, whose pollen data and age models have been homogenised and updated. Records outside the homogenised set are temporarily extracted to check the 40 pre-existing age models and recalculate them if necessary, using their radiocarbon data available on Neotoma. Models were generated on R Studio (Team, 2020) with the Bacon package (Blaauw and Christen, 2011, version 3.3.0) using the IntCal20 calibration (Reimer et al., 45 2020). A minimum of 4 radiocarbon dates was set, in line with the study by Li et al. (2022), to allow the preexisting age-depth model to be updated. Records for which this constraint was unmet were removed from the fossil pollen dataset.
- Temporal extent filter: For the two zones, pollen records must cover the period before 8,000 cal. BP and after 5,000 cal. BP and its signal must be continuous between

**Figure 2.** (a) Location of the 38 fossil pollen records used for palaeoclimate reconstruction. Current mean (b) summer and (c) winter temperature (top) and precipitation (bottom) repartition in the Mediterranean basin. Current climate parameters are based on average monthly climate data for 1970-2000 and extracted from WorldClim version 2.1 (Fick and Hijmans, 2017).

8,000–5,000 cal. BP. The application of this constraint is intended to guarantee that any climate trend or oscillation that occurs can be documented.

 Resolution filter: only records with a satisfactory mean temporal resolution were kept, i.e., inferior to 350 years.sample<sup>-1</sup>.

For the pollen data, the Neotoma data were retained because pollen homogenisation of the LegacyPollen database results in a loss of information due to the level of group-10 ing of taxa, making the use of these homogenised pollen records impractical for the application of our multi-method approach to reconstructing climate change. Aquatic taxa and spores were excluded, and rare herbaceous and woody taxa were grouped at higher taxonomic levels (e.g., genus, sub-15 family, or family). For each record, the local environmental context was checked against the original publication (a time-consuming but necessary step) to highlight the presence of local hygrophilous and/or mesohygrophilous taxa (e.g., Cyperaceae, Alnus glutinosa-type, Salix, Populus). In certain 20 records, it was necessary to eliminate these "local" taxa more characteristic of wetlands, such as shallow lakes and peat bogs (1 record in southern France and 11 records in Italy).

Ultimately, 38 records were selected: 17 from southern France and 22 from Italy (16 from Neotoma, 6 provided by authors: Sadori and Narcisi 2001; Mercuri et al. 2002; Joannin et al. 2012, 2013; Mercuri et al. 2013; Joannin et al. 2014b; Sadori 2018; d'Oliveira et al. 2023). Only two required full age—depth model recalibration.

### 2.3 Modern pollen data

30 Different studies underline the importance of the modern pollen and climate datasets used for palaeoclimate reconstruction (Turner et al., 2021) and some of them point out the advantage of regional datasets (Dugerdil et al., 2021a, b). To test the role of the modern dataset on the reconstruction (Trasune et al., 2024), and improve the reliability of 35 our climate reconstructions, three modern pollen datasets were used, one global and two regional. The Eurasian Pollen Dataset (EAPDB), compiled by Peyron et al. (2013, 2017) and updated by Dugerdil et al. (2021a) and Robles et al. (2022, 2023), was used as the global modern pollen dataset. 40 The EAPDB contains a total of 3,373 pollen surface samples (Fig. A1a). From the EAPDB dataset, we derived two regional modern datasets by sub-sampling the EAPDB. The two regional datasets correspond to a temperate pollen dataset (TEMPDB) and a Mediterranean pollen dataset 45 (MEDDB). For the TEMPDB, samples from the EAPDB were extracted using both a spatial selection (Western Europe) and a temperate biome selection (warm mixed forest, xerophytic wood/shrub, temperate deciduous forest, cool mixed forest, warm steppe, cold mixed forest, and cool 50 steppe). Similarly to the TEMPDB, the MEDDB is the result of resampling the EAPDB through spatial selection (the Mediterranean basin) and a selection by biomes (warm mixed forest, xerophytic wood/shrub, temperate deciduous forest, warm steppe, cold mixed forest, and cool mixed forest). After compiling the two regional datasets, the TEMPDB and the MEDDB contain 1,875 and 1,040 surface pollen samples respectively (Fig. A1b-c). For the three modern pollen datasets (EAPDB, MEDDB and TEMPDB), a total of 103 taxa were used. Taxa representing less than 0.1 % of 60 the pollen spectra at a given site were removed.

For each modern pollen sample, seasonal and annual climate values were extracted from the WorldClim2.1 dataset (Fick and Hijmans, 2017) as follows: Mean annual temperature (MAAT), spring temperature (Tspr), summer temperature (Tsum), autumn temperature (Taut), winter temperature (Twin), mean annual precipitation (MAP), spring precipita-

**Table 1.** Fossil pollen records used for the climatic reconstruction. Records are classified by zone (1: southern France; 2: Italy) and by a west–east gradient for zone 1 and by a north–south gradient for zone 2.

| ID | Sitename                    | Latitude<br>(° N) | Longitude<br>(° E) | Elevation (m) | Zone | Age min<br>(yrs BP) | Age max<br>(yrs BP) | Resolution<br>(yrs.sple <sup>-1</sup> ) | Modern<br>dataset | Database | Reference                                                             |  |
|----|-----------------------------|-------------------|--------------------|---------------|------|---------------------|---------------------|-----------------------------------------|-------------------|----------|-----------------------------------------------------------------------|--|
| 1  | Planell de Perafita         | 42.48             | 1.57               | 2231          | 1    | 3091                | 10158               | 157.04                                  | EAPDB             | Neotoma  | Miras et al. (2010)                                                   |  |
| 2  | Bosc dels Estanyons         | 42.48             | 1.63               | 2296          | 1    | -55                 | 11899               | 129.93                                  | EAPDB             | Neotoma  | De Beaulieu et al. (2005);<br>Miras et al. (2007)                     |  |
| 3  | Lake Racou                  | 42.55             | 2.01               | 2014          | 1    | 284                 | 12148               | 152.1                                   | EAPDB             | Neotoma  | Guiter et al. (2005)                                                  |  |
| 4  | Les Palanques               | 42.16             | 2.44               | 465           | 1    | 39                  | 8491                | 130.03                                  | TEMPDB            | Neotoma  | Pérez-Obiol (1988);<br>Piqué et al. (2018);<br>Revelles et al. (2018) |  |
| 5  | Canroute                    | 43.65             | 2.58               | 790           | 1    | -77                 | 12494               | 241.75                                  | TEMPDB            | -        | d'Oliveira et al. (2023)                                              |  |
| 6  | Peyre peat-bog              | 44.96             | 2.72               | 1097          | 1    | 308                 | 12453               | 94.15                                   | TEMPDB            | Neotoma  | Surmely et al. (2009)                                                 |  |
| 7  | Brameloup                   | 44.74             | 3.08               | 1224          | 1    | 72                  | 12499               | 146.2                                   | TEMPDB            | Neotoma  | De Beaulieu et al. (1985)                                             |  |
| 8  | Bonnecombe                  | 44.57             | 3.13               | 1388          | 1    | 1698                | 12423               | 214.5                                   | TEMPDB            | Neotoma  | De Beaulieu et al. (1985)                                             |  |
| 9  | Tourbiere des Narses Mortes | 44.43             | 3.6                | 1256          | 1    | -20                 | 10671               | 334.09                                  | TEMPDB            | Neotoma  | Beaulieu (1974)                                                       |  |
|    |                             |                   |                    |               |      |                     |                     |                                         |                   |          | Reille and De Beaulieu (1988);                                        |  |
| 10 | Lac du Bouchet              | 44.92             | 3.78               | 1181          | 1    | -47                 | 12324               | 268.93                                  | TEMPDB            | Neotoma  | Thouseny et al. (1990)                                                |  |
| 11 | Embouchac                   | 43.57             | 3.92               | 9             | 1    | 1461                | 9417                | 42.32                                   | TEMPDB            | Neotoma  | Puertas (1998, 1999)                                                  |  |
| 12 | Tourves                     | 43.41             | 5.91               | 304           | 1    | 1582                | 16035               | 123.53                                  | TEMPDB            | Neotoma  | Nicol-Pichard (1987)                                                  |  |
| 13 | Correo                      | 44.56             | 6                  | 1101          | 1    | 662                 | 12496               | 81.61                                   | TEMPDB            | Neotoma  | Nakagawa (1998)                                                       |  |
| 14 | Vallon de Provence          | 44.39             | 6.4                | 2075          | 1    | 2891                | 11809               | 107.45                                  | EAPDB             | Neotoma  | De Beaulieu and Jorda (1977)                                          |  |
|    |                             |                   |                    |               |      |                     |                     |                                         |                   |          | De Beaulieu and Jorda (1977);                                         |  |
| 15 | Lac de Siguret              | 44.61             | 6.56               | 1066          | 1    | 3879                | 12408               | 213.23                                  | TEMPDB            | Neotoma  | Beaulieu and Reille (1983)                                            |  |
| 16 | Plan du Laus                | 44.24             | 6.7                | 1783          | 1    | 922                 | 11978               | 155.72                                  | EAPDB             | Neotoma  | De Beaulieu and Jorda (1977)                                          |  |
| 17 | Lac des Grenouilles         | 44.1              | 7.48               | 1810          | 1    | 2177                | 12159               | 133.09                                  | EAPDB             | Neotoma  | Finsinger et al. (2021)                                               |  |
| 18 | Schwarzsee                  | 46.67             | 11.43              | 2033          | 2    | 34                  | 11600               | 131.43                                  | EAPDB             | Neotoma  | Seiwald (1979)                                                        |  |
| 19 | Malschotscher Hotter        | 46.67             | 11.46              | 2050          | 2    | 0                   | 11492               | 208.95                                  | EAPDB             | Neotoma  | Seiwald (1979)                                                        |  |
| 20 | Dura Moor                   | 46.64             | 11.46              | 2080          | 2    | 7                   | 12461               | 125.8                                   | EAPDB             | Neotoma  | Seiwald (1979)                                                        |  |
| 21 | Rinderplatz                 | 46.64             | 11.49              | 1780          | 2    | -7                  | 12374               | 113.59                                  | EAPDB             | Neotoma  | Seiwald (1979)                                                        |  |
| 22 | Balladrum                   | 46.16             | 8.75               | 390           | 2    | -30                 | 12358               | 176.97                                  | TEMPDB            | Neotoma  | Hofstetter et al. (2006)                                              |  |
| 23 | Lago di Ledro               | 45.87             | 10.75              | 652           | 2    | 10                  | 18156               | 92.11                                   | TEMPDB            | -        | Joannin et al. (2013, 2014b)                                          |  |
| 24 | Lago del Segrino            | 45.83             | 9.26               | 374           | 2    | -19                 | 12166               | 133.9                                   | TEMPDB            | Neotoma  | Gobet et al. (2000)                                                   |  |
|    |                             |                   |                    |               |      |                     |                     |                                         |                   |          | Finsinger and Tinner (2006);                                          |  |
| 25 | Lago Piccolo di Avigliana   | 45.05             | 7.39               | 356           | 2    | 331                 | 12478               | 36.59                                   | TEMPDB            | Neotoma  | Finsinger et al. (2011)                                               |  |
| 26 | Pavullo nel Frignano        | 44.32             | 10.84              | 675           | 2    | 50                  | 12341               | 279.34                                  | TEMPDB            | Neotoma  | Vescovi et al. (2010a)                                                |  |
| 27 | Lago Padule                 | 44.3              | 10.21              | 1187          | 2    | -10                 | 11834               | 171.65                                  | TEMPDB            | Neotoma  | Watson (1996)                                                         |  |
| 28 | Lago del Greppo             | 44.12             | 10.67              | 1442          | 2    | -60                 | 11031               | 96.44                                   | TEMPDB            | Neotoma  | Vescovi et al. (2010b)                                                |  |
| 29 | Lago dell'Accesa            | 42.99             | 10.9               | 157           | 2    | 261                 | 11842               | 87.73                                   | MEDDB             | Neotoma  | Drescher-Schneider et al. (2007)                                      |  |
| 30 | Lago di Mezzano             | 42.36             | 11.46              | 452           | 2    | 67                  | 15309               | 127.02                                  | TEMPDB            | -        | Sadori (2018)                                                         |  |
| 31 | Lago di Martignano          | 42.11             | 12.32              | 200           | 2    | 83                  | 12599               | 184.06                                  | TEMPDB            | Neotoma  | Kelly and Huntley (1991)                                              |  |
| 32 | Lago di Nemi                | 41.72             | 12.7               | 320           | 2    | 3.1                 | 11386               | 130.84                                  | MEDDB             | -        | Mercuri et al. (2002, 2013)                                           |  |
| 33 | Lago Grande di Monticchio   | 40.93             | 15.61              | 656           | 2    | 87                  | 12436               | 138.75                                  | TEMPDB            | Neotoma  | Allen and Huntley (2018)                                              |  |
| 34 | Lago Trifoglietti           | 39.55             | 16.02              | 1048          | 2    | 14                  | 11439               | 69.24                                   | TEMPDB            | -        | Joannin et al. (2012)                                                 |  |
| 35 | Urio Quattrocchi            | 37.9              | 14.4               | 1044          | 2    | 3154                | 10348               | 78.2                                    | MEDDB             | Neotoma  | Bisculm et al. (2012)                                                 |  |
| 36 | Lago Preola                 | 37.62             | 12.63              | 6             | 2    | -46                 | 10366               | 160.18                                  | MEDDB             | Neotoma  | Calò et al. (2012)                                                    |  |
| 37 | Gorgo Basso                 | 37.6              | 12.65              | 6             | 2    | -57                 | 10501               | 121.36                                  | MEDDB             | Neotoma  | Tinner et al. (2009)                                                  |  |
| 38 | Lago di Pergusa             | 37.52             | 14.3               | 667           | 2    | 53                  | 12749               | 156.74                                  | MEDDB             | -        | Sadori and Narcisi (2001)                                             |  |

tion (Pspr), summer precipitation (Psum), autumn precipitation (Paut) and winter precipitation (Pwin).

The modern dataset used for each fossil record was adapted according to the dominant taxa, and regional datasets (TEMPDB and MEDDB) were used whenever possible. The MEDDB was selected when at least 15% of Mediterranean taxa (*Olea, Phillyrea, Pistacia* and *Quercus ilex*-type) were present in the record; otherwise, the TEMPDB was selected. Where certain mountainous/boreal taxa such as *Picea* or *Be*-10 tula were dominant (≥ 75%) the global EAPDB dataset was selected to limit the no-analogue situations. To facilitate the identification of climatic trends, the records were organised along a longitudinal west–east gradient for southern France and a north–south latitudinal gradient for Italy. The potential biases associated with the elevation of the sites (Ortu et al., 2006) were also investigated, and the fossil records have been

distinguished into two categories: low (< 1000 m) and high (> 1000 m) elevations.

# 2.4 A multi-method approach to reconstruct past climate from pollen data

Over the last decades, several methods have been developed to reconstruct climate from pollen data (see review by Chevalier et al., 2020). As these methods are based on different ecological concepts and mathematical algorithms, results can be strongly method-dependent (Chevalier et al., 2020). In this frame, multi-method approaches have been developed to increase the reliability of the results (Peyron et al., 2005, 2011, 2013, 2017; Salonen et al., 2019; Robles et al., 2022, 2023; Sassoon et al., 2025). These methods were initially developed to calibrate the relationship between modern pollen data (soils, mosses) and current climate parameters. Multi-method approaches include (1) assemblages ap-

proach, based on the principle of dissimilarity between fossil and modern assemblages (Modern Analogue Technique); (2) transfer functions, based on linear or non-linear regressions (Weighted Averaging Partial-Least Squares regression) between pollen taxa and climate parameters, and (3) recent machine learning techniques with regression trees (Random Forest and Boosted Regression Trees) to quantify climate parameters.

The Modern Analogue Technique (MAT; Guiot, 1990) is widely used to reconstruct past climates due to its straightforward application, efficacy, and sensitivity. This technique relies on evaluating the dissimilarity between each fossil and modern pollen assemblages and selecting the closest modern samples (known as analogues). The WA-PLS method, introduced by ter Braak and Juggins (1993), operates as a transfer function that assumes an unimodal relationship between the proportions of pollen and climatic conditions. It suggests that the abundance of a species is intrinsically linked to its environmental tolerance. WA-PLS calculates the climatic optimum for a species based on calibration data by determining the average climatic conditions in which the species is found, weighted according to its abundance (Chevalier et al., 2020).

The two additional methodologies, namely Random Forest (RF) and Boosted Regression Trees (BRT), have emerged 25 more recently and are grounded in Machine Learning principles (Salonen et al., 2019). These techniques utilise regression trees to systematically partition pollen data through successive divisions based on the abundance observed in the pollen spectrum. The Random Forest approach relies on the 30 estimation and aggregation of multiple regression trees, with each tree being derived from a collection of pollen samples using a bootstrapping technique (Chevalier et al., 2020). In contrast, Boosted Regression Trees differ from RF in their treatment of the modern dataset: while RF assigns equal se-35 lection probabilities to all samples, BRT increases the likelihood of selecting under-represented samples from the previous tree. This technique, known as boosting, enhances the model's predictive accuracy for elements that are less accurately predicted (Salonen et al., 2019). Here, the final output 40 of the BRT method is derived from averaging the results of 15 independent executions of the BRT algorithm, reflecting the variability inherent in regression tree signals.

For each method, the reliability of the results was estimated by bootstrap cross-validation by calculating the values of the correlation coefficient between the variables (R<sup>2</sup>) and those of the root mean square error criterion (RMSE).

### 2.5 Numerical analyses

The relationships between climate variables were assessed with a scatter-plot matrix to provide an overview of the distributions and correlations of the variables, and a pair-plot was drawn.

To assess the relationships between modern pollen spectra and the climatic variables, an ordination technique was used on each modern pollen dataset. Major pollen taxa (those present in a least 15 samples and with a maximum  $\geq 3\%$ ) of each dataset were square-root-transformed to stabilise variances and optimise the signal-to-noise ratio (Prentice, 1980). A detrended correspondence analysis (DCA; Hill and Gauch, 1980) was applied to each modern pollen dataset to determine whether a linear-based analysis or unimodal-based analysis was more appropriate based on gradient length as the criterion. The DCA results showed that, for the three modern datasets, the gradient length was over 3.0 standard deviation, suggesting that unimodal-based methods should be used in further analyses on the modern pollen datasets (Ter Braak and Verdonschot, 1995).

A canonical correspondence analysis (CCA) was carried out to detect the influence of climate variables on each modern pollen dataset, and variance inflation factors (VIFs) were also used to determine the correlations between environmental variables. VIF values > 10 indicate co-linearity with other variables, which may lead to unreliable results (Ter Braak and Prentice, 1988; Cao et al., 2014). To eliminate the correlation between the variables, we selected the temperature and precipitation variables with a large explained variance and eliminated those with a VIF > 10.

Climate reconstructions based on different methods and reliability tests (R<sup>2</sup> criterion and RMSE) were performed with the packages rioja (Juggins, 2024, version 1.0.7), randomForest (Breiman, 2001, version 4.7-1.2) and dismo (Hijmans et al., 2024, version 1.3.16). All analyses were performed on R Studio (Team, 2020, version 4.4.1), using the ggplot2 package (Wickham, 2016, version 3.5.1) for graph creation.

To facilitate the comparison of climate signals between each record and the identification of potential patterns of changing climate trends between records, a normalisation between -1 and 1 (rescaling min-max) is applied to the smoothing of the mean of selected models for each record. Data normalisation is based on the following equation Eq. (1):

$$x^{'} = a + \frac{(x - min(x))}{(max(x) - min(x))} \times (b - a)$$
 (1)

With x non-smoothed initial mean values,  $x^{'}$  normalised smoothed values, a and b minimal and maximal normalisation values, in this case, respectively -1 and 1.

Because normalisation is based on the overall mean of the record, any interval influenced by human activity is deliberately excluded from the standardisation process. This precaution prevents the representation from being skewed toward artificially drier or warmer values. Determining the boundaries of human impact relies either on author notes from the original fossil record publications or on detecting a significant abundance of anthropogenic indicator taxa such as: Olea, Juglans, Castanea, Cerealia-type, Plantago lanceolata, and Rumex-type.

Composite curves are constructed for the central Mediter- 105 ranean area and will be compared to other palaeoclimate syn-

theses of the Mediterranean region (Cruz-Silva et al., 2023; Liu et al., 2023). For the composite curves, non-normalised reconstruction results were used, without the exclusion of periods where the human activity is discernible. Reconstructed 5 palaeoclimate values of every climate parameter were averaged in a 300 year bin, the maximal resolution of all records being 306 years. For each record, the first bin was centred on 0 years BP, and the following bins were centred on a 300 year increment throughout the record. Then the binned 10 values of each record were averaged to produce a regional climate signal for the central Mediterranean. Finally, the regional climate signal values were transformed into anomalies. The anomaly calculation is based on the reconstructed modern average value. The estimation of confidence inter-15 vals for each composite, encompassing the 5th and 95th percentiles, was achieved through bootstrap resampling at the site level, utilising 1000 iterations.

#### 3 Results

#### 3.1 Ordinations

20 The pair-plots of the ten climatic variables for all three modern pollen datasets (Fig. B1) indicate that mean annual temperature (MAAT) and precipitation (MAP) are highly correlated (Pearson correlation coefficient > 0.7) with the seasonal parameters (Taut, Tspr, Twin, Tsum and Paut, Pspr, Pwin, 25 Psum). Among seasonal parameters, spring (Tspr and Pspr) and autumn (Taut and Paut) parameters are the ones which are also highly correlated between them. This co-linearity is supported by the canonical correspondence analyses (CCAs) of the pollen assemblages and the climate variables for the 30 three modern datasets (Fig. C1). For each modern pollen dataset, the first CCA, with every climate parameter, indicated that the variance inflation factor (VIF) values of annual (TANN, PANN), spring (Tspr and Pspr) and autumn (Taut and Paut) parameters are greater than 10 (Fig. C1a-c). After 35 deleting those parameters, the remaining four climate variables (Tsum, Twin, Psum and Pwin) have VIF values lower than 10 and, therefore, can be used in the final CCA to investigate their influence on modern pollen datasets (Fig. C1d-f).

### 3.2 Model performances

<sup>40</sup> Results of model performances, estimated by bootstrap cross-validation, are summarised in Fig. 3 with the correlation coefficient (R<sup>2</sup>, Fig. 3a) and the root mean square error criterion (RMSE, Fig. 3b1b2). For all climate parameters and all modern datasets, best R<sup>2</sup> and RMSE values, i.e., re<sup>45</sup> spectively highest and lowest values, are obtained for MAT and BRT methods. Conversely, the WA-PLS and RF methods show lower R<sup>2</sup> and RMSE values. Therefore, we will retain here the MAT and BRT methods for the interpretation and discussion of our climate reconstructions.

**Figure 3.** Performance results of the four methods tested (MAT, WA-PLS, BRT, and RF) with the three different modern pollen datasets – the modern Eurasian (EAPDB), Temperate (TEMPDB), and Mediterranean (MEDDB) datasets – for summer and winter precipitation (Psum, Pwin) and temperature (Tsum, Twin). (a) R-squared values (R<sup>2</sup>). (b1) Root Mean Square Error of precipitation (RMSE). (b2) Root Mean Square Error of temperatures (RMSE).

### 3.3 Quantitative palaeoclimate reconstructions

The MAT and BRT were applied to each fossil pollen record to reconstruct the summer and winter climate parameters (Figs. 4 and 5). To compare more easily the different sequences, the results obtained with both methods were averaged for each record. This averaged signal was then normalised (rescaling min-max method, Eq. (1)) according to the mean value of the smoothed climatic signal for the entire record.

50

100

### 3.3.1 Summer conditions: temperature and precipitation

Southern France was influenced by a west-east temperature gradient (Fig. 4a). The western and central regions of zone 1 (records 1 to 11) are characterised by a downward trend in 5 summer temperatures during the Holocene. Tsum were relatively high from the beginning of the Holocene to around 6,000 years cal. BP before falling until the modern period. This climatic pattern, particularly evidenced in the high elevation records, suggests the presence of a Holocene Thermal 10 Maximum (HTM) in these regions. For the more easterly regions of southern France (records 12 to 17), an opposite pattern appears to be present, with higher summer temperature trends toward the most recent period. In Italy, summer temperatures seem to be strongly influenced by a north-south 15 division with different patterns, distributed on either side of latitude 43° N. The region of zone 2 located above 43° N (records 18 to 28) is characterised by two different patterns depending on high or low elevation (Fig. 4a). At high elevations (records 18 to 21 and 27 to 28), high summer tem-20 perature values are reconstructed from the beginning of the Holocene to around 6,000 years cal. BP before falling until the recent period. This pattern is similar to the one observed at high elevations in southern France, also suggesting the presence of an HTM in the Italian Alps. At lower ele-25 vations (records 22 to 26), low summer temperature values are evidenced from the beginning of the Holocene to around 5,000 years cal. BP, followed by a summer warming to the modern-day period. Below 43° N, an opposite climatic pattern to that at low elevation is present, corresponding to rel-30 atively low Tsum values at the beginning of the Holocene, followed by an increase from 8,000 years cal. BP onward, which continues into modern times.

For summer precipitation (Psum), three patterns, depending on a west-east gradient, are evidenced in southern France 35 (Fig. 4b). In the westernmost region of zone 1 (records 1 to 4), high precipitation is reconstructed between the early and mid Holocene periods (around 9,000-4,500 years cal. BP). A decrease in summer precipitation occurs during the late Holocene, more or less early depending on the record, 40 suggesting summer aridification throughout the Holocene. In contrast, records from the central region of southern France (records 5 to 11) indicate dry conditions during the early Holocene, followed by a wetter mid to late Holocene. Highelevation sequences highlight a precipitation maximum ear-45 lier and longer (around 8,000-4,000 years cal. BP) than the low-elevation sequences (6,000–3,000 years cal. BP). Further east (records 11 to 17), we reconstruct a pattern similar to that observed for the western regions of southern France (high precipitation followed by an aridification) but with 50 an earlier and shorter humid period, i.e., wetter conditions present during the early Holocene period (12,000-8,000 years cal. BP) before aridification from the onset of mid Holocene onward. In Italy, similarly to summer temperatures, summer rainfall is strongly influenced by a north-south gradient. Contrasting climatic patterns are highlighted on ei- 55 ther side of latitude 43° N. Above 43° N, low precipitation is reconstructed during the early Holocene until the onset of the mid Holocene (12,000-8,000 years cal. BP), followed by high precipitation throughout the mid-to-late Holocene. Wetter summers are evidenced between 7,500 and 4,500 years 60 cal. BP. In contrast, below 43° N, an opposite precipitation trend is reconstructed, with wet summers during the early Holocene (with a maximum around 9,000 years cal. BP), followed by drier conditions throughout the Holocene (particularly evidenced after 7,000 years cal. BP).

### 3.3.2 Winter conditions: temperature and precipitation

In zone 1, two distinct patterns are reconstructed (Twin, Fig. 5a). In the western and eastern regions of southern France (records 1 to 3 and 12 to 17), cold winter conditions occurred at the beginning of the Holocene, followed by warmer conditions. In the west (records 1 to 3), warmest conditions occur from the middle Holocene (8,000 years cal. BP) onward. In the east (records 12 to 17), the warming happens earlier and seems to last longer, from the early Holocene to the mid-tolate Holocene transition periods (around 11,000–5,000 years cal. BP) depending on the record. The central region of southern France (records 5 to 11), on the other hand, shows a more contrasted climatic signal, characterised by warm conditions during the early Holocene, followed by a temperature decrease during the mid Holocene before an increase 80 until the modern period. In Italy, the winter climate signal seems much less influenced by the north-south division than in summer. The contrasting conditions observed in summer around 43° N are not depicted here; the climate signal is more homogeneous. Most of Italy is characterised by cold condi-85 tions during the early Holocene, followed by a gradual increase in winter temperatures throughout the Holocene.

For precipitation trends, we observed a spatial repartition similar to winter temperatures in southern France (Fig. 5b). The eastern and western regions of zone 1 (records 1 to 3 and 12 to 17) are defined by the same trends, i.e., an increase in winter precipitation over the Holocene, with wetter conditions observed from the mid Holocene period (8,000 years cal. BP) onward. In the central regions of southern France (records 5 to 11), Fig. 5 also suggests a winter precipitation 95 increase although (1) the climatic signal seems more contrasted between the records, (2) the wetter conditions are observed later, during the mid-to-late Holocene transition period (between 5,000-3,000 years cal. BP), than in the other two regions of zone 1.

In the Alpine part of Italy and the northern Apennines (records 18 to 28), a similar pattern to that observed in the eastern part of southern France is evidenced, i.e., an increase in winter precipitation during the Holocene. However, the onset of wetter winter conditions seems to have been de- 105 layed at high altitudes (records 18 to 21). At lower elevations (records 23 to 25), the wet conditions occur earlier, during

**Figure 4.** Summer (a) temperature (Tsum) and (b) precipitation (Psum) reconstructed from 38 fossil pollen records. The colour gradient corresponds to models-averaged min-max normalised values of each climate parameter. Positive (negative) values correspond to summer conditions that are (a) warmer (colder) and (b) wetter (drier) than the mean value of the climate signal for each record.

the early Holocene period (around 10,000 years cal. BP). In central Italy, in the regions of southern Tuscany and Lazio (records 29 to 32), there appears to be a general trend in winter throughout the Holocene, similar to that observed in northern Italy, i.e., a gradual increase in precipitation over the Holocene, with a stronger temporal heterogeneity between records. In Sicily (records 35 to 38), dry conditions are evidenced at the start of the Holocene until around 8,000 years cal. BP, followed by wetter conditions during the middle Holocene period (around 7,000 years cal. BP, depending on the record).

#### 4 Discussion

We used a pollen-inferred multi-method approach (3 modern datasets, 2 methods) to reconstruct specifically the climate changes in the central Mediterranean area over the last 12,000 years BP, with a focus on the seasonality estimates (temperature and precipitation of both winter and summer conditions). Results suggest that local/regional trends differ in the central Mediterranean during the Holocene. Summer conditions were characterised by a west—east distinction in southern France and a north—south one in Italy, for both temperature and precipitation. The HTM in summer has been found north of 43° N at high elevation (≥ 1000 m).

In contrast, Holocene winter conditions showed a more homogeneous spatio-temporal pattern, i.e., general wetter and warmer conditions throughout the Holocene in both Italy and southern France, although some local discrepancies can be evidenced depending on the elevation of the site.

### 4.1 Importance of chronological considerations

We discuss here the chronology range, quality and resolution for each record, which are particularly important as they may induce some biases in the interpretation of the climatic reconstructions. For this study, specific criteria were applied to select records with a fairly good chronology, i.e., a continuous temporal range between 10,000 and 5,000 years cal. BP and an average temporal resolution of less than 350 years.sample<sup>-1</sup>. However, it should be pointed out that the quality of the chronologies is not equal between each fossil pollen record, which may initially affect the accuracy of the reconstructions, but also the interpretation that will be made of them and can be considered as an unavoidable limitation in our reconstructions.

Another chronological concern is linked to the partitioning of Holocene climatic trends. Certain climatic periods that are emblematic of the Holocene in Europe, such as the HTM, are not always identified at the same time and/or with the same intensity, for reasons that may depend on chronology,

**Figure 5.** Winter (a) temperature (Twin) and (b) precipitation (Pwin) inferred from the 38 fossil pollen records. The colour gradient corresponds to models-averaged min-max normalised values of each climate parameter. Positive (negative) values correspond to winter conditions that are (a) warmer (colder) and (b) wetter (drier) than the mean value of the climate signal for each record.

proxy sensitivity (marine vs terrestrial) and/or local response to climatic change (e.g., elevation, Bini et al., 2019). This is why the time interval definition is not always a simple or obvious choice. Divisions specific to the Mediterranean region have been proposed, notably by Jalut et al. (2000) who proposed a division into three periods with (1) a lower humid Holocene (11,500–7,000 years cal. BP), (2) a transition phase (7,000–5,000 years cal. BP) and (3) an upper Holocene (5,500 years cal. BP—present) characterised by aridification.

In contrast to the aforementioned study and to better discuss the climate changes of the central Mediterranean area in terms of spatio-temporal patterns, we choose to divide the Holocene into four periods: (1) 12,000–8,000 years cal. BP corresponding to the Lateglacial and early Holocene, (2) 8,000–5,000 years cal. BP during which the HTM should be observable if recorded, (3) 5,000–3,000 years cal. BP associated with post-HTM conditions and (4) 3,000 years cal. BP–present during which the human impact on vegetation is recorded in most of fossil pollen records.

### Pollen-inferred uncertainties to be taken into account for Holocene palaeoclimate reconstructions in the Mediterranean region

Among the uncertainties that can impact our palaeoclimatic reconstructions, it seems essential to address the limitations of the pollen proxy when used in palaeoclimatic quantifica- 25 tion. One of the first uncertainties that can be stated is the effect of the agglomeration of several archive types (e.g., lakes and peat bogs) with different local recording conditions (e.g., sediment accumulation rate, edaphic conditions, hydroseral succession). Variability in spatial representativeness can af- 30 fect the reconstructed palaeoclimatic signal, e.g., a lake associated with a small catchment will record a more local signal than a lake associated with a large catchment, which would better represent regional vegetation (Sugita, 1993; Herzschuh et al., 2023b). A second factor, a migrational lag, can impact 35 our reconstructions based on pollen data, as vegetation can take time to return to equilibrium with its environment after a major and rapid climate change (Mauri et al., 2015). The impact of the migrational lag is generally considered for the Lateglacial-early Holocene transition, which corresponds to 40 the last rapid climate shift of the last 12,000 years BP, when most of the postglacial vegetation began to take hold and before climate changes became less strong and less rapid. These uncertainties could impact the reconstructions of the Lateglacial and early Holocene periods of our study. Finally, 45 it seems essential to address the impact of human presence and its interaction with vegetation when pollen data are used to reconstruct palaeoclimates. Several studies have already shown that human activity, through land clearance for grazing or cultivation, can lead to an over-representation of nonarboreal pollen, and thus influence palaeoclimate reconstructions in which vegetation changes do not entirely correspond to climatic variations (Birks et al., 2010; Mauri et al., 2015).

In the central Mediterranean, studies showed that the main cause of vegetation changes before 4,000 years BP was climatic variations, but after, from the late Holocene onward, vegetation changes seemed to be attributed to human activity and climatic influences altogether (Sadori et al., 2011; Marignani et al., 2017). For our study, the modification of vegetation by humans is a point to be taken into account for palaeoclimate interpretations since the presence of anthropogenic taxa (i.e., Olea, Juglans, Castanea, Cerealia-type, Plantago lanceolata and Rumex-type) is observed in several records of 5 zones 1 and 2 from around 3,000 years BP onward.

Studies have highlighted the importance of site effects on the response to climate change, such as complex topography and elevation, which can act as important perturbation factors to large-scale atmospheric flows (Beniston et al., 1997). 20 The study by Ortu et al. (2010), focused on the Alps, showed that differences in the pollen assemblages of high ( $\geq 1000$ m) and low (< 1000 m) elevation sites play a significant role in the quality of climate reconstructions based on pollen. In our study, elevation also seems to play a role, as the HTM 25 observed in the south of France and the Italian Alps is best recorded at high elevations north of 43° N. However, this interpretation may be questioned because (1) most pollen located north of 43° N is located at high elevations (19 out of 28 records) and (2) high elevation vegetation in the Alps may 30 be mostly controlled by summer temperature while low elevation vegetation in southern Italy may be more controlled by water availability and/or precipitation. A greater number of high-elevation records south of 43° N would enable us to refine our climate interpretation, particularly in the south of 35 France, where continuous pollen records are lacking.

## 4.3 A north–south climate division of the Italian Peninsula

The pioneering study by Magny et al. (2012, 2013) first highlighted a north-south palaeohydrological contrast in the cen-40 tral Mediterranean during the Holocene. In addition, they identified a latitudinal "tipping point" around 40° N (Magny et al., 2013), splitting and distinguishing two distinct and opposite patterns on either side of latitude 40° N. In contrast to the study of Magny et al. (2013), which is based on 6 records, 45 with only one record located between latitude 40 and 45° N, our synthesis is based on 38 records, 22 of which are located along the Italian Peninsula, giving us a unique opportunity to see whether the same north-south division also appears in our results at a wider scale. Here, a latitudinal division on 50 either side of 43° N in Italy throughout the Holocene is evidenced in summer while winter climate conditions appear to have been more spatially homogeneous and rather be marked by a stronger temporal dynamic (Figs. 4, 5, D1, E1). Our study corroborates the findings of Magny et al. (2013) but suggests that the "tipping point" takes place at higher latitudes, around 43° N, and only impacts summer climate conditions (Figs. 4, D1). Through a multi-proxy study, Robles et al. (2023) also highlighted a north–south climate division in Italy during the Lateglacial period, proposing a threshold around latitude 42° N. Our study aligns with this pattern and suggests that the location of the latitudinal threshold dividing the Italian climate may have varied slightly through time and space.

During the early Holocene period (12,000–8,000 years cal. BP), the south of 43° N is characterised by relatively cold 65 and wet summer conditions, regardless of elevation, while the north of 43° N is rather associated with hot and dry ones, particularly at high elevation (Figs. 4, D1), which is in line with the observations made by Magny et al. (2012) at this period. Winter conditions were, however, rather cold and dry throughout the early Holocene period (Figs. 5, E1), in contrast to what was suggested by the aforementioned study. The mid Holocene period (8,000-5,000 years cal. BP) was associated with relatively warm and humid winter conditions in the main part of Italy. Warmer and drier summers are re- 75 constructed south of 43° N, in contrast to wetter summers recorded north of 43° N. This pattern for the mid Holocene differs from the Magny et al. (2013) study, which reported humid winters with dry summers north of 40° N and humid winters and summers south of 40° N during the mid 80 Holocene. From the mid-to-late Holocene transition onward (5,000 years cal. BP to modern-day), winter and summer climate trends south of 43° N are globally similar to the mid Holocene ones, while colder climate conditions following the HTM are depicted in summer from the sites located north of 85 43° N at high elevation (Fig. 4a).

In the Mediterranean, the seasonal and spatial variability of precipitation is directly linked to the global atmospheric circulation (Trigo and DaCamara, 2000). In their study, Magny et al. (2013) highlighted the combined effects of the blocking of the North Atlantic anticyclone linked to variations in summer insolation and the influence of ice sheets and the forcing of freshwater melt in the north Atlantic ocean, both of which have an impact on large-scale circulation processes such as the NAO. In addition to global atmospheric 95 circulation, other more local factors, such as orography, latitude and oceanic and/or continental influences, are directly linked to the seasonal and spatial variability of precipitation. The intricate topography of central Italy, combined with its central location within the Mediterranean basin, leads to 100 precipitation patterns that arise from a variety of meteorological processes and influences, including flows from the south-west, west, north-east, and south-east (Silvestri et al., 2022). This complexity complicates the relationship between precipitation variability and alterations in large-scale atmo- 105 spheric circulation, rendering the connections among precipitation fluxes, orography, and the spatial distribution of rainfall more intricate than in other regions characterised by simpler precipitation mechanisms (Silvestri et al., 2022), such as those predominantly influenced by westerly winds, as seen in Southern California (Millán et al., 2005; Hughes et al., 2009).

### 5 4.4 A west–east climate gradient in the central Mediterranean

The north-south climate contrast does not appear to be the only specific climate dynamic in the Mediterranean basin. Roberts et al. (2011, 2012) have shown in a synthesis on 10 the mid Holocene climate transition that both proxy data and model outputs suggested a west-east division in the Mediterranean climate history. Specifically, western Mediterranean early Holocene changes in precipitation were significantly smaller in magnitude and spatially less coherent than the 15 eastern ones, and the mid Holocene (around 6,000-3,000 years cal. BP) recorded the rainfall maximum before a decline to present-day values. Studies suggest that these contrasting hydroclimate regimes indicate a complicated interaction between different atmospheric systems, e.g., the main 20 phases of NAO-like circulation (Fletcher et al., 2013; Di Rita et al., 2018), the East Atlantic pattern and the size and position of the North African anticyclone, expressed differently in various sectors of the Mediterranean (Joannin et al., 2014b; Di Rita et al., 2018). This longitudinal component 25 of the Mediterranean climate is also found in our study, which contrast the western regions of southern France with the northern Italian regions (Figs. 4b and 5b), and placing the central zone of southern France in a zone where the climatic signal is much more contrasted, suggesting the pres-30 ence of a "buffer zone" where the different atmospheric dynamics confront each other. However, this hypothesis merits a more detailed regional study to understand the complexity of this transition zone, located in the southern part of the Massif Central region where relatively high elevation are en-35 countered (> 600 m) and several climatic influences come together, i.e., Mediterranean influence from the south; the influence of the Atlantic Ocean from the west due to Atlantic air masses arriving from the country's west coast, which are not prevented by any geographical barriers obstacles in the 40 Aquitaine Basin; and a mountainous regime from the north.

# 4.5 Spatio-temporal climate trends in the Mediterranean area during the Holocene

Our climate reconstruction, based on 38 records located in the central Mediterranean, highlights reliable spatio-45 temporal patterns at a regional scale despite an expected variability across the records that could reflect differences in latitude, elevation or in situ characteristics. We thus reconstruct homogeneous winter conditions in contrast to summers, which were characterised by a marked north–south re-50 versal throughout the Holocene (Figs. 4, 5).

### 4.5.1 Data-model comparison

Our results have been compared with the GCMs transient simulations TraCE-21ka (Collins et al., 2006). Comparison between data and model simulations will give us a better understanding of the climate mechanisms and their forcings. As the forcings implemented in TraCE-21ka are known (i.e., changes in orbital configuration and atmospheric greenhouse gases, the extent, topography and changing palaeogeography of ice sheets and scenario of freshwater forcing to the oceans from the retreating ice sheets), consistency or inconsistencies between the reconstructions and the simulations will enable us to highlight the forcings involved in the observed climate variations.

Pollen data and TraCE-21ka simulations show a winter warming throughout the Holocene (Fig. 7b-d). Discrepan- 65 cies are evidenced in the precipitation estimates for the early Holocene period (pollen: dry conditions, TraCE-21ka: wet conditions), but models and data are in relatively good agreement for the mid-to-late Holocene (Fig. 6b-e). The range of anomaly values based on models (-19.0 to 18.8 mm) is lower 70 than the values derived from pollen data (-79.3 to 30.4 mm). TraCE model simulations indicate a gradual summer humidification during the Holocene, although with a more accentuated humidification for regions north of 43° N than for those south of 43° N (Fig. 6d), which contrasts with the pollenbased summer aridification south of 43° N (Fig. 6a). The model simulations with TraCE-21ka (Figs. 6d, 7a) do not evidence the north-south latitudinal patterns depicted by the pollen climate reconstruction. Indeed, the simulated trends are consistent between the north of 43° N and the south of 80 43° N, with colder summer conditions at the beginning of the Holocene, followed by a warming, and a thermal maximum from 8,000 to 5,000 years cal. BP, then a gradual cooling. The thermal maximum during the mid Holocene period is more pronounced in regions located north of 43° N. Such a 85 pattern is consistent with other model simulations for northern Europe and follows the change of summer insolation at those latitudes, which peaked at the onset of the Holocene (Mauri et al., 2015; Russo and Cubasch, 2016).

The data—model discrepancies may be explained by known biases in the spatial distribution of continental precipitation of the CCSM3 simulations (Collins et al., 2006) and by the coarse spatial resolution (≥ 0.5°) of GCMs as CCMS3 (Karger et al., 2023). Regional models will be more helpful to simulate the climate conditions during the Holocene at the Mediterranean scale (Brayshaw et al., 2011; Peyron et al., 2017) as they can reproduce realistic climatology with respect to the observations (Russo and Cubasch, 2016; Russo et al., 2022).

# 4.5.2 Winter: a warming trend in the Mediterranean region

Our pollen-inferred climate reconstruction indicates wet conditions on either side of 43° N in the central Mediterranean from 6,000 years cal. BP (Fig. 6b). The timing depends on the latitude: the wetter conditions are recorded around 7,000 years cal. BP at the north of 43° N, and later around 5,000 years cal. BP at the south of 43° N followed by a gradual increase until present-day values.

For temperatures, a rapid warming is reconstructed from 12,000 to 9,000 years cal. BP followed by a temperature increase throughout the Holocene, on either side of 43° N (Fig. 7d).

One of our goals was to study the climate changes in 15 the central Mediterranean to fill the gap between the Iberian Peninsula (Fig. 7f) and Eastern Mediterranean (Fig. 7h) regions. Those two regions display the same winter warming trend throughout the Holocene associated with negative anomalies, similar to our central Mediterranean winter tem-20 perature reconstructions. Winter warming associated with positive winter anomalies from 8,500 years BP onward are present in the composite curves of Kaufman et al. (2020a) encompassing regions between the 30°-, , 60° N parallels. This may suggest that the winter warming magnitude may 25 vary regionally. The winter warming trend of the Mediterranean basin, inferred by proxy data and model simulations, are similar and follows the variations in winter insolation for these latitudes, i.e., increasing trend, suggesting that insolation plays an important role and may be one of the main forc-30 ing of temperature variations in winter during the Holocene (Cruz-Silva et al., 2023; Liu et al., 2023).

# 4.5.3 Summer: a contrasted spatio-temporal trend in the Mediterranean region

The summer trends are more contrasted than the winter ones <sup>35</sup> and depend strongly on the latitude. While regions south of <sup>43°</sup> N are characterised by a progressive decrease of summer precipitation throughout the Holocene, regions north of <sup>43°</sup> N are characterised by dry conditions from 12,000 to 7,000 years cal. BP, followed by a mid Holocene wetness maxi-<sup>40</sup> mum (7,000–5,000 years cal. BP), before the gradual onset of summer aridification (Fig. 6a).

For summer temperatures, the results also seem strongly linked to latitude. This is illustrated with the opposed climate trends north and south of 43° N (Fig. 7c). While the north of 43° N signal shows the presence of a summer thermal maximum (HTM) between 10,000–6,000 years cal. BP followed by a gradual cooling, regions located south of 43° N experienced a warming trend from 11,000 years cal. BP onward with warmer conditions during the late Holocene than during the mid Holocene (Fig. 7c). Similar to our northern signal, the study of Zhang et al. (2022) highlighted an evident seasonal difference in southeastern Europe from pollen

data, with a slightly warmer-than-present summer, associated with an early HTM between 10,000 and 8,000 years BP, but a cooler-than-present winter during the early Holocene before modern-day temperatures are reached. Zhang et al. (2022) explained this seasonal difference by orbital-induced seasonal insolation changes, playing a critical role in seasonal temperature evolutions, which in a long-term decrease in temperature seasonality during the Holocene. Our results are also coherent with the study of Marriner et al. (2022), which highlighted the presence of an HTM in the Mediterranean Sea from marine biomarkers.

This study provides new results on the well-debated question of whether southern Europe was colder than the north 65 during the mid Holocene. Our results show that the mid Holocene was indeed colder than the late Holocene in the southern zone, while in the northern parts warmer conditions, associated with an HTM, were present. This should be nuanced, however, by the fact that the anomalies were calcu-70 lated here in relation to the more recent reconstructed values (0-300 years binned) and not according to measured modern values, which could bias the anomaly calculation. Nevertheless, warmer or close to present-day summer temperature anomalies were present in the eastern, central and western Mediterranean (Fig. 7c-e-g-i) during the mid Holocene, questioning previous observations from pollen-based reconstructions showing colder temperatures in the Mediterranean region around 6,000 BP (Cheddadi et al., 1997; Davis et al., 2003; Mauri et al., 2015). Previous studies that depicted 80 colder summer temperature in the Mediterranean around the mid Holocene used a similar pollen-based reconstruction method, i.e., Plant Functional Type (PFT), Modern Analogue Technique (MAT), while our study and the ones depicting a relatively close to modern-day summer temperature dur- 85 ing the mid Holocene (Cruz-Silva et al., 2023; Herzschuh et al., 2023a; Liu et al., 2023) used different methods, i.e., a multi-method approach combining the MAT and BRT or a version of the Weighted-Partial Least Squared (WA-PLS) method. This suggests that a method bias may be present, 90 overestimating the cold values reconstructed in summer for the Mediterranean with the MAT method used by previous studies. The use of different methods producing similar results supports our observations and highlights the value of multi-method approaches for future studies based on pollen 95 data to reconstruct palaeoclimates.

At a larger scale, we observe similar trends between the southern central Mediterranean (South 43° N) and the Eastern Mediterranean (Cruz-Silva et al., 2023, Fig. 7g), with relatively warm conditions at the onset of the Holocene, followed by a cooling around 11,000 years cal. BP, and then by a gradual warming from 11,000 to 4,000 years cal. BP. This trend is not evidenced either on the Iberian peninsula or in the Herzschuh et al. (2023a) study (orange and coral lines respectively on Fig. 7g), focusing on southern Europe (between latitudes 40–50° N and 30–40° N). On the other hand, even though the signal is slightly different, the Herzschuh

**Figure 6.** Reconstruction of the Holocene composite signal of the mean values across records for this study (north-central Mediterranean), using 300 years as the bin, expressed as anomalies relative to the current reconstructed values of summer and winter precipitation (mm) from (a-b-c) the pollen-based signal and (d-e-f) the TraCE-21ka model-based signal. Total (convective and large-scale) precipitation rate (PRECT) was used to extract seasonal (summer and winter) precipitation simulations from the atmosphere post-processed data containing decadal mean seasonal averages. Output from the simulation was extracted for each pollen record location (Tab. 1). Composite curves were then constructed following the same process as pollen-based climate reconstruction. Shaded area corresponds to standard deviation values through a bootstrap resampling at the site level utilising 1000 iterations.

et al. (2023a) study also clearly shows a different trend between the sites located north and south of 40° N, with a more pronounced optimum for the more northerly sites, which corroborates our results. In the Herzschuh et al. (2023a) study, 5 some of the pollen records used to reconstruct the summer conditions for the 40-50° N area (orange line on Fig. 7i) are shared with our study, but a lot of their fossil records are located in the Alps, central and northern France, southern Germany and Bulgaria, regions with distinct climate dynamics 10 compared to the Mediterranean ones, particularly in summer as the Mediterranean climate is known for its marked seasonality (Xoplaki et al., 2003). For the southern sites, few pollen records used in the Herzschuh et al. (2023a) study to reconstruct the 30–40° N climate signal (coral line on Fig. 15 7i) are shared with our study, and most of them are located in the southern Iberian Peninsula, in Greece and Anatolia. Modern-day climate dynamics are different in the western and eastern Mediterranean compared to the central region (Lionello et al., 2006), which may explain the differences observed between our reconstructions and the one proposed by Herzschuh et al. (2023a). This can explain some differences together with the method used, i.e., WA-PLS vs. the combined MAT-BRT multi-method approach, and the modern pollen dataset used, i.e., the global modern dataset encompassing 15,379 sites over Eurasia and North America vs. 25 the regional modern dataset.

However, factors other than methodology differences may explain the discrepancies obtained for the summer season. Indeed, the winter temperature signal reconstructed for the Iberian Peninsula, Eastern Mediterranean and the central Mediterranean is consistent between the studies, indicating that differences in methodology may not be the most important factor in explaining the summer climate trends observed. Another explanation could be the presence of a marked west–east Mediterranean climatic gradient (Roberts et al., 35 2011; Finné et al., 2019), probably present before the be-

**Figure 7.** Reconstruction of the Holocene composite signal of the mean values across records for this study (north-central Mediterranean), using 300 years as the bin, expressed as anomalies relative to the current reconstructed values of summer and winter temperatures (°C) from (a-b) the pollen-based signal and (c-d) the TraCE-21ka model-based signal. Surface temperature (TS) was used to extract seasonal (summer and winter) temperature simulations from the atmosphere post-processed data containing decadal mean seasonal averages. Output from the simulation was extracted for each pollen record location (Tab. 1). Composite curves were then constructed following the same process as pollen-based climate reconstruction. From (e) to (g), the reconstructed signal of summer temperatures for (e) the Iberian Peninsula (digitalised from Liu et al., 2023), (f) Eastern Mediterranean (Cruz-Silva et al., 2023) and (g) southern Europe (Herzschuh et al., 2023a). From (h) to (i), reconstructed signal of winter temperatures for (h) the Iberian Peninsula (digitalised from Liu et al., 2023), and (i) Eastern Mediterranean (Cruz-Silva et al., 2023). Shaded area corresponds to standard deviation values through a bootstrap resampling at the site level utilising 1000 iterations.

ginning of the Holocene, impacting the summer period and making the spatio-temporal pattern of summer climatic conditions much more complex than in winter.

# 4.5.4 Central Mediterranean seasonality evolution during the Holocene

The latitudinal component of the Mediterranean climate is also reflected in changes in precipitation seasonality. North

of 43° N, summer and winter precipitation show similar increasing trends, although more pronounced in winter than in summer (Fig. 6c1). Regions south of 43° N show a different seasonal pattern, with increasingly arid summers and in-5 creasingly wet winters (Fig. 6c2). This "Mediterraneanization" phenomenon of the Mediterranean basin took place during the mid Holocene after 8,000 years BP and seems to have had a greater impact on the southern regions than on the northern regions (Joannin et al., 2014a; Finné et al., 10 2019). The latitudinal variability of "Mediterraneanization" is also observed by Sadori et al. (2011), who showed that the expansion of drought-resistant taxa around the Mediterranean basin, which occurred principally after 8,000 years BP and was associated with a re-organisation of regional cli-15 mate, was more pronounced in the southern regions of the central Mediterranean (e.g., Spain, Sicily, Croatia and southern Greece) than in central and northern Italy and the inner parts of the Balkans.

#### 5 Conclusions

<sup>20</sup> We aimed to document the climate changes in the central Mediterranean during the Holocene, including trends and different patterns. A robust methodology has been applied to 38 pollen records spreading across the south of France and Italy. Four climate reconstruction methods based on different mathematical and ecological concepts have been tested (MAT, WA-PLS, BRT and RF), and the selection of the best modern calibration dataset has also been investigated to produce the most reliable results. Particular attention has been paid to the seasonal nature of climatic parameters (winter and summer temperatures and precipitation). A data-model comparison has been made using transient model simulation TraCE-21ka in an attempt to gain a better understanding of the climate mechanisms and their forcing.

Our palaeoclimate reconstruction shows that:

(1) During the mid Holocene, summer temperatures were slightly colder but close to present-day conditions in the southern part of the central Mediterranean region. This is coherent with the summer temperature reconstructions of the Iberian peninsula, eastern Mediterranean, and at a more re-40 gional scale of southern Europe between 30° N to 50° N for the mid Holocene, showing anomalies close to modernday values between 8,000 and 6,000 years BP. In northern parts of the central Mediterranean region, and particularly in high elevation (≥ 1000 m), a Holocene thermal maximum is 45 present. Those observations contrast with the cold summer temperature anomalies previously reconstructed with pollen data for the Mediterranean region. We suggest that a method bias may be responsible for an overestimation of cold summer temperature, highlighting the benefit of multi-method 50 approaches, which could reduce the biases expressed by the use of a single method.

- (2) Holocene summer conditions were characterised by specific spatio-temporal patterns, i.e., a west-east differentiation in southern France and a north-south one in Italy, for both temperature and precipitation. This latitudinal division 55 on either side of 43° N for summer conditions confirms the initial hypotheses exposed by Magny et al. (2012, 2013) and Peyron et al. (2017), which correlated it with the decline of the possible blocking effects of the North Atlantic anticyclone linked to maximum insolation and of the influence of 60 the remnant ice sheets and freshwater forcing in the North Atlantic ocean. Holocene winter conditions showed a more homogeneous spatio-temporal pattern, i.e., general humidification and warming throughout the Holocene for Italy and southern France. This spatial homogeneity of the winter cli- 65 mate throughout the Holocene is coherent with other pollenbased studies focusing on the Eastern Mediterranean and the Iberian Peninsula. Both longitudinal, latitudinal and altitudinal dynamics define the spatio-temporal summer variability. Winter conditions in the north Mediterranean are much 70 less affected by spatial characteristics and follow the increase in winter insolation at those latitudes. The presence of a west-east pattern, opposing the climate of western southern France and the north of Italy, supports previous observations opposing the north-western and the south-eastern Mediterranean, highlighting the presence of complicated interaction between different atmospheric systems.
- (3) Our pollen-based reconstructions with the TraCE-21ka transient simulations are mostly incoherent, particularly for precipitation, which may be explained by the presence of sys-80 tematic biases in the spatial distribution of continental precipitation of CCSM3 simulations and by a still too coarse spatial resolution. The complex orography of the region, with the presence of the Alps, the Pyrenees and the Apennines, combined with the too-coarse spatial resolution of 85 global climate models (GCMs), prevent a good simulation of the spatial distribution of precipitation. Those discrepancies between model simulations and pollen-based reconstructions also suggest that during the Holocene, the northern Mediterranean climate was already subject to a marked 90 spatio-temporal variability, particularly in summer, that cannot only be explained by changes in orbital configuration and atmospheric greenhouse gas evolution.
- (4) Our result highlighted the onset of the "Mediterraneanization" of the central Mediterranean region, characterised by wet winters and dry summers, after 8,000 years BP. The "Mediterraneanization" process seems to have had a greater impact on the southern regions than on the northern regions.

This study has enabled us to further complete our un- 100 derstanding of the spatio-temporal variability of the north-central Mediterranean, in particular by comparing reconstructions based on similar methods applied to a corpus of pollen records. However, the limitations of this proxy are well known, although difficult to control and quantify, 105 and the differences between reconstructions based on in-

dependent proxies have been highlighted on several occasions. This is why multi-proxy approaches, applied to several records, would make it possible to strengthen the robustness of our palaeoclimatic reconstructions based on pollen data. Several independent proxies have already been used in previous studies, such as chironomids for summer temperatures, oxygen isotopes for precipitation, and lipid biomarkers (e.g., alkenones and brGDGTs) for temperatures and environmental indicators. However, these studies remain relatively poor in the Mediterranean, highlighting the need to carry out these multi-proxy approaches on new sequences.

### Appendix A

**Figure A1.** Location of surface sites used in (a) the Eurasian Pollen Dataset (EAPDB) compiled by (Peyron et al., 2013, 2017), (b) the Temperate Dataset (TEMPDB) and (c) the Mediterranean Dataset (MEDDB).

### Appendix B

**Figure B1.** Pair-plots of the ten climatic variables (MAAT, MAP, Tsum, Twin Taut, Tspr, Psum, Pwin, Paut and Pspr) for (a) the Eurasian Pollen Dataset (EAPDB), (b) the Temperate Dataset (TEMPDB) and (c) the Mediterranean Dataset (MEDDB).

### Appendix C

**Figure C1.** Canonical Correspondence Analysis (CCA) results and Variance Inflation Factor (VIF) values for (a and d) the Eurasian Pollen Database (EAPDB), (b and e) the Temperate Pollen Database (TEMPDB) and (c and f) the Mediterranean Pollen Database (MEDDB). The variance of axis 1 (CCA1) and axis 2 (CCA2) is expressed in percentages on the axis label. Panels (a) to (c) correspond to (top) the first canonical correlation analysis (CCAs) and (bottom) VIF values applied to all climate variables. Panels (d) to (f) correspond to (top) the second canonical correlation analysis and (bottom) VIF values without climate variables with VIF values > 10. Dashed lines mark the threshold of 10 used to on VIF values determine the presence of correlation between the environmental variables.

### Appendix D

### **Summer**

**Figure D1.** Maps of Holocene precipitation and temperature changes during the four designed periods of this study for summer conditions, where each record is represented by a disc divided into two parts with the precipitation information on one side (purple or yellow) and the temperature one on the other side (red or blue). The climate signal has been averaged through each period to propose an average trend for the period under consideration. The horizontal red line corresponds to the 43° N latitudinal delimitation characteristic of the north–south division of the Holocene Mediterranean climate.

### Appendix E

37.5

**Figure E1.** Maps of Holocene precipitation and temperature changes during the four designed periods of this study for winter conditions, where each record is represented by a disc divided into two parts with the precipitation information on one side (purple or yellow) and the temperature one on the other side (red or blue). The climate signal has been averaged through each period to propose an average trend for the period under consideration. The horizontal red line corresponds to the 43° N latitudinal delimitation characteristic of the north–south division of the Holocene Mediterranean climate.

Longitude (°E)

### Appendix F

**Table F1.** Variance inflation factors (VIFs) statistics including ten climate parameters for each of the modern datasets. VIF values < 10 are highlighted in bold. Ticks correspond to the absence of the climate parameter in the multicollinearity calculation.

|         | MAAT      | MAP    | Tsum     | Twin     | Taut     | Tspr     | Psum  | Pwin  | Paut  | Pspr  |
|---------|-----------|--------|----------|----------|----------|----------|-------|-------|-------|-------|
|         | 544965.62 | 127.00 | 15063.97 | 78116.43 | 36146.23 | 34022.81 | 18.11 | 36.44 | 24.24 | _     |
|         | -         | 126.99 | 21.57    | 80.50    | 135.19   | 46.42    | 18.08 | 36.40 | 24.22 | -     |
| EAPDB   | -         | -      | 21.57    | 80.50    | 135.19   | 46.42    | 5.09  | 9.35  | 9.76  | 7.52  |
| EAPDD   | -         | -      | 14.42    | 21.67    | -        | 43.0 9   | 4.91  | 9.33  | 9.52  | 7.29  |
|         | -         | -      | 2.60     | 2.86     | -        | -        | 3.22  | 9.24  | 9.03  | 7.10  |
|         | -         | -      | 2.56     | 2.45     | -        | -        | 1.68  | 1.60  | -     | -     |
|         | 170519.23 | 157.66 | 10379.25 | 13520.43 | 11691.39 | 10409.15 | 30.71 | 38.89 | 23.67 | -     |
|         | -         | 157.64 | 49.64    | 48.33    | 107.88   | 60.95    | 30.69 | 38.88 | 23.62 | -     |
| TEMPDB  | -         | -      | 49.64    | 48.33    | 107.88   | 60.95    | 8.91  | 7.78  | 8.66  | 7.46  |
| LEMITUD | -         | -      | 29.74    | 18.55    | -        | 60.78    | 8.64  | 7.24  | 8.27  | 7.45  |
|         | -         | -      | 4.39     | 3.95     | -        | -        | 4.46  | 6.85  | 7.07  | 6.50  |
|         | -         | -      | 3.87     | 3.73     | -        | -        | 2.76  | 1.45  | -     | -     |
|         | 184683.87 | 192.85 | 11657.98 | 12169.64 | 11859.77 | 12569.80 | 30.29 | 48.19 | 31.48 | -     |
|         | -         | 192.27 | 35.99    | 36.59    | 112.87   | 76.50    | 30.29 | 48.16 | 31.28 | -     |
| MEDDB   | -         | -      | 35.99    | 36.59    | 112.87   | 76.50    | 9.66  | 7.06  | 9.03  | 11.69 |
| MEDDD   | -         | -      | 27.53    | 20.32    | -        | 65.92    | 9.66  | 5.69  | 8.62  | 11.05 |
|         | -         | -      | 5.96     | 4.87     | -        | -        | 8.19  | 5.65  | 7.66  | 11.02 |
|         | -         | -      | 5.80     | 4.59     | -        | -        | 3.12  | 1.21  | -     | -     |

Data availability. A part of the pollen data used is publicly accessible on the Neotoma Paleoecology Database (http://www.neotomadb.org). Access to pollen data retrieved directly from the authors of the original studies should be requested 5 directly from the authors. Palaeoclimate reconstruction will be fully available on PANGAEA.

Author contributions. Ld'O performed the analytical work, and LD designed the R codes; Ld'O, SJ, NC-N, MB, and OP designed the study; SJ, GM, NC-N, and OP supervised the study; AF, AM, 10 AMM and LS provided part of the study material (fossil pollen sequences); LD, MB and MR contributed to data analysis; MB provided financial support for the project; Ld'O wrote the manuscript draft; SJ, GM, NC-N, LD, MB, MR, AF, AM, AMM, LS, MB and OP reviewed and edited the manuscript.

- 15 Competing interests. At least one of the (co-)authors (O. Peyron) is a member of the editorial board of Climate of the Past but the authors declare that they have no known competing financial interests or personal relationships that could have appeared to influence the work reported in this paper.
- <sup>20</sup> **Acknowledgements.** This research was funded, in whole or in part, by [ANR AUTUMN-LAMBS, Grant ANR-22-CE27-0011]. A CC-BY public copyright license has been applied by the authors to the present document and will be applied to all subsequent versions up to the Author Accepted Manuscript arising from this submission, in accordance with the grant's open access conditions. Data were obtained from the Neotoma Paleoecology Database (http://www.neotomadb.org) and its constituent database(s), see Table 1. Conference funding was provided by the Association des Palynologues de Langue Française (APLF). The work of data contributors, data stewards, and the Neotoma community is gratefully acknowledged. This is ISEM contribution ISEM 2025-144.

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
