# Peer review of "Holocene Climate Dynamics in the Central Mediterranean Inferred from Pollen Data"

_EGUsphere, 2025_

## Author Response (AR1)

Final response for the paper "Palaeoclimate synthesis of the central Mediterranean area from pollen data" by L. d'Oliveira et al.

**Egusphere-2025-1106**

We thank both reviewers for their thorough reading and constructive comments. As suggested by both reviewers, we streamlined the manuscript to make it clearer, more concise, and more easy to read. With the same intent, Figs. A3 and A4 remain in the appendix section, while Fig. 3 has also been moved there. Finally, thanks to the comment of Reviewer #2, we corrected an analytical step in our method, leading to an improvement in our results and the modification of two figures (Figs. 7 and 8), which do not change our principal conclusions but increase the robustness of our approach.

**Reviewer #1**

D'Oliveira and co-authors present a data synthesis of Holocene pollen records from the Mediterranean. With this dataset, the authors test different climate transfer functions to best reconstruct seasonal changes in temperature and precipitation and then compare with transient climate model simulations to assess the spatio-temporal trends and their forcing mechanisms. I found the synthesis, data analysis and model comparisons to be robust with reasonable conclusions drawn from them and only have a few comments regarding those areas. Figures are great although I have suggestions for additional ones that I hope will better showcase their large effort. I also have some suggestions for text that I also hope will streamline their manuscript. Overall, this manuscript will be a valuable addition to the literature and fits well within *Climate of the Past*'s scope.

**General Comments**

While naturally lengthy given the large dataset studied, I found the manuscript to be long and not focused in parts. I think the text could benefit from being stream-lined in areas. For example, the introduction contains some short paragraphs (e.g., L80-84) that could be folded in elsewhere. In general, a paragraph should be about 5-7 sentences, ideally with a topic and concluding sentence that help tie it to previous and subsequent paragraphs. Similarly in the Study Area section (2.1), the NAO description (L124-131) could easily, and more logically, be combined with the preceding paragraph that discusses it. Another example is the Fossil Pollen Data Section (2.2.1). There is a lot of good information in here but could be restructured to be clearer. I would keep the first paragraph and then each following one should be focused specifically on the selection criteria. For instance, age models are "updated" early on but only explained at the end and should be discussed together. Better yet, perhaps this could be presented as bullets?

I encourage the authors to carefully review the entire manuscript in this light to consolidate it as much as possible. It will only help the reader better understand the considerable (and great) work you did!

**Response:** We are grateful for these comments and we agree that some parts of the manuscripts could be combined and better synthesised. The parts highlighted above will all be reworked. In an echo to Reviewer #2, a particular effort will be made to streamline all the manuscript.

Figures 5 and 6 are really nice, and I think helps distil the spatio-temporal patterns of climate change across the region. I may also suggest the addition of some spatial maps that would be helpful for explaining both proxy and model-based results (e.g., McKay et al., 2024, Nat Comm, https://doi.org/10.1038/s41467-024-50886-w). These could be perhaps a difference between early and mid-Holocene conditions and/or discrete time slices. I think it would be a nice way to visualize all your results.

**Response:** We agree with this suggestion and have documented the spatio-temporal variability of our results in two figures in the appendices. These two appendices represent the spatial dispersion of the evolution of summer (Fig. A3) and winter (Fig. A4) temperatures and precipitation for each fossil record for four time

periods: 12,000-9,000 cal BP; 9,000-6,000 cal BP; 6,000-3,000 cal BP, and 3,000-0 cal BP. In particular, this representation highlights the spatially homogenous evolution of winter conditions, the presence of a summer thermal maximum with mostly high elevations, and the completeness of the summer signal. In an intent to keep the manuscript clear and to further streamline the manuscript, also in response to Reviewer #2's comments, Figs. A3 and A4 will remain in the appendix section.

**Specific Comments**

L6: can't say "none of them" as the two exceptions you highlight clearly do - rephrase to none have focused on the "entire" Med?

Text changes: "none of them" corrected as "few of them".

L12 (and throughout): precipitations should be singular precipitation.

**Text changes:** "precipitations" corrected to "precipitation" throughout the manuscript.

L14: were colder – colder than what? Please clarify.

**Text changes:** Sentence modified to "summer temperatures were colder than modern-day conditions [...]"

L22: A data-model comparison "shows"

**Text changes:** Corrected to "shows".

L31: Define BP;

Text changes: acronym "BP" defined as "Before Present (BP)".

L35: remove "well"

Response: OK.

L39: when were these conditions? Early or mid-Holocene? Please clarify.

**Text changes:** Sentence modified to "terrestrial proxies suggest conditions similar to or cooler than those observed today during the mid-Holocene period [...]" for more clarity.

L70: Chironomid should be singular.

**Text changes:** "Chironomids" corrected to "Chironomid".

L76: "giving good robustness" is awkward, please consider rephrasing. In addition, this paragraph could benefit from some brief discussion on the limitations of pollen climate reconstructions as elaborated on latter in the manuscript.

**Response:** We acknowledge the awkwardness of this sentence and rephrase it. Further discussion on this paragraph will be added in the discussion section 4.2, addressing pollen uncertainties to be taken into account for palaeoclimate reconstructions.

**Text changes:** Sentence rephrased as "supporting our confidence in the interpretation of palaeoclimatic reconstructions derived from pollen data".

L88: You never really discuss marine proxies in the discussion. Please add this in later on or consider removing here.

**Response:** The presence of an HTM in marine proxies will be discussed later in the manuscript.

L97: Please acknowledge what this single method is for reference.

**Text changes:** "such as Modern Analogue Technique (MAT) or Weighted Averaged Partial Squared (WA-PLS) methods" will be added for more clarity.

L33 (and throughout): I would suggest removing "very"...it doesn't mean much of anything quantitatively. Response: The term "very" will be deleted or replaced by another formulation that is more relevant and quantitatively informative, depending on the context.

L160-161: It seems conflicting that at first the span must cover 8000 year BP, but then 10000 year BP. Please clarify why these are different or edit to align.

**Response:** The text will be edited as such "the pollen record must cover the period before 8,000 years cal. BP and after 5,000 years cal. BP."

L184: and climate datasets used "for paleoclimate reconstructs?" please clarify what the climate datasets are used for here.

**Response:** The sentence will be modified to "Different studies underline the importance of the modern pollen and climate datasets used for palaeoclimate reconstruction (Turner et al., 2021) [...]" to clarify the idea.

L190: From the EAPDB, we derive two smaller datasets...It is not explicitly clear from the current text that YOU did this. It's a nice contribution and should be acknowledged as such!

**Response:** We are grateful for the acknowledgement of our work and contribution. We will modify the aforementioned sentence as proposed to underline our contribution and change the text as follow: "From the EAPDB dataset, we derived two regional modern datasets by sub-sampling the EAPDB. The two regional datasets correspond to a temperate pollen dataset (TEMPDB) and a Mediterranean pollen dataset (MEDDB).

L390: Uncertainties should be plural.

**Text changes:** Corrected to "uncertainties".

L405: attributed should be past tense.

**Text changes:** Corrected to "attributed".

L418: remove "has".

Response: OK.

L508: goals should be plural.

Response: OK.

L513: induced isn't quite the right word, perhaps inferred?

**Text changes:** "induced" corrected to "inferred"

L590: Start point (1) as a new paragraph to maintain consistency with following numbered concluding points.

**Response:** "Our palaeoclimate reconstruction shows that (1) [...]" paragraph structure will be modified to "Our palaeoclimate reconstruction shows that:

(1) [...]"

**Reviewer #2**

The paper summarizes the results of quantitative climate reconstructions from the central Mediterranean area using fossil pollen data from 38 Holocene pollen records and two quantitative climate reconstruction techniques (MAT and BRT). The study is in general thorough, resulting in a number of detailed figures and a longish text. It is good that the authors have included for example ordination analyses such as DCA and CCA for exploring their data, but I would recommend relocating Fig. 3 in the supplement to streamline the paper.

**Response:** We thank Reviewer #2 for his comment and suggestion. Fig. 3 will be relocated in the appendix section to further streamline the manuscript, also in response to Reviewer #1's comments.

The results of the study, shown in Figs. 5-8, reflect the remarkable regional and temporal variability, which characterizes both temperature and precipitation records in summer and in winter. As a result, it is not easy to find what the key results of the study are, unless the variability itself is the main result. Such a distinct site-to-site variability with no clear Holocene trends is a bit worrying and makes one wonder whether the reasons for it may be in the complexity of the climate factors driving the Holocene vegetation patterns in the study region. For example, it is possible that some of the high-altitude sites in the Alps have been mostly controlled by summer temperature, while the sites in southern Italy more by water availability/precipitation.

**Response:** Variability from one site to another exists, particularly for summer conditions. This may reflect complex regional climatic variability, but also the presence of potential site effects. As the aim of this study is to reconstruct a regional signal, local particularities have not been explored in depth in this work, although we recognise their undeniable importance. Nevertheless, clear trends can be observed from one sub-region to another, highlighting a clear north-south division in summer and a homogenous spatio-temporal winter evolution.

We agree with Reviewer #2 on the physiological factor of vegetation on the reconstructed climatic parameters and their interpretations. A discussion will be added on this subject, in particular on the observation of the thermal maximum at high altitudes in summer and its absence at low altitudes. Text will be modified accordingly.

My most important comment concerns the timescales of the work, especially the Late Holocene. We can see in Figs. 5-6 that the 38 normalized records do not generally speaking cover the last 1000-3000 years. The reason for this is explained on page 12 "To not bias this representation toward drier or warmer values, the period for which the impact of human activity is discernible in the pollen diagrams, have been excluded from the standardization process". This decision, of course, places a high importance on how the period of human activity is defined. The resulting graphs in Figs. 5-6 are thus floating sequences, where the temperature and precipitation trends over the last 2000-3000 years are absent. Most palaeoclimatologists would certainly appreciate seeing the sequences reaching up to the present, instead of the floating sequences.

Response: We thank Reviewer #2 for this comment. We consider that human impact on vegetation is an important factor to take into account in the interpretation of the results of the palaeoclimate reconstructions based on pollen data, for recent time periods. As pointed out by the reviewer, we have chosen not to include in our standardisation process the time periods for which human impact is recorded by vegetation. We stress that for the last 2000-3000 years, when human presence is attested in the Mediterranean region, the climatic factor on vegetation remains present, but it becomes difficult to distinguish it from the human factor. This is why we are taking great care when interpreting our results for these periods. We do not believe that it is relevant to include periods when the pollen signal is marked by the presence of humans in the normalisation calculation, which would bias the normalised values and therefore our interpretations. Similarly, if these periods are excluded from the normalisation but still represented, there is a very high risk of values greater than 1, since the human footprint tends to overestimate the reconstructed temperatures, which would confuse the interpretation of the results.

However, we recognise that this is an important point which will be better explained in the new version of the manuscript.

The summary climate curves in Fig. 8 are different, because in these curves the periods of human activity have not been excluded. The resulting curves are very confusing. They indicate that the summer temperature curves both south and north of 43° remain steady until a dramatic dip of c. 2.0 °C over the last 1000 years. A similar but weaker decline can be seen in the winter temperature (Fig. 8 b). If this was true, it would indicate that the central Mediterranean has undergone a dramatic cooling over the recent centuries. As far as I know,

this is against all historical palaeoclimatic information from the Mediterranean region and it is surprising that the authors do not discuss this distinct feature in their data. I suspect that this drop is fully spurious, maybe caused by the strong human impact. In any case, it weakens the feasibility of the reconstructions and cannot be passed without any comments or discussion in the paper.

**Response:** We would sincerely like to thank Reviewer #2 for his comment on the spectacular drop of 2.0°C over the last 1000 years and for questioning it. This led us to check carefully our results, our aggregation methods, and our choice of representation. This "spectacular anomaly" was a result of how the periods with missing data were treated in the script. Our first approach only took account for continuous signal during our time windows of 300 BP. Missing values in record for this time window (e.g., continuous signal from 150 to 300 cal. BP and not from 0 to 300 cal. BP) would therefore not be considered in the records aggregation, which would give an erroneous value that is not representative of our results. We have corrected it in our aggregation method and the results do not show the spurious drop in temperature and precipitation anymore. We propose new Figs. 7 and 8 that can be seen below. The strong drop was not caused by climate and/or human factors; therefore, no further discussion will be added on this subject in the manuscript.

The correction of an analytical step in our method lead to an improvement of our results. They do not change our principal conclusions and interpretation but increase the robustness of our approach.

**Figures rebuttal:**

**Figure 7.** Reconstruction of the Holocene composite signal of the mean values across records for this study (north-central Mediterranean), using 300 years as the bin, expressed as anomalies relative to the current reconstructed values of respectively summer and winter precipitation (mm) from (a-b-c) the pollen-based signal and (d-e-f) the TraCE-21ka model-based signal. Total (convective and large-scale) precipitation rate (PRECT) was used to extract seasonal (summer and winter) precipitation simulations from the atmosphere post-processed data containing decadal mean seasonal averages. Output from the simulation was extracted for each pollen record location (Tab. 1). Composite curves were then constructed following the same process

as pollen-based climate reconstruction. Shaded area corresponds to standard deviation values through a bootstrap resampling at the site level utilising 1000 iterations.

**Figure 8.** Reconstruction of the Holocene composite signal of the mean values across records for this study (north-central Mediterranean), using 300 years as the bin, expressed as anomalies relative to the current reconstructed values of respectively summer and winter temperatures (°C) from (a-b) the pollen-based signal and (c-d) the TraCE-21ka model-based signal. Surface temperature (TS) was used to extract seasonal (summer and winter) temperature simulations from the atmosphere post-processed data containing decadal mean seasonal averages. Output from the simulation was extracted for each pollen record location (Tab. 1). Composite curves were then constructed following the same process as pollen-based climate reconstruction. From (e) to (g), the reconstructed signal of summer temperatures for (e) the Iberian Peninsula (digitalised from Liu et al., 2023), (f) Eastern Mediterranean (Cruz-Silva et al., 2023) and (g) southern Europe (Herzschuh et al., 2023a). From (h) to (i), reconstructed signal of winter temperatures for (h) the Iberian Peninsula (digitalised from Liu et al., 2023), and (i) Eastern Mediterranean (Cruz-Silva et al., 2023). Shaded area corresponds to standard deviation values through a bootstrap resampling at the site level utilising 1000 iterations.